# Inhibition of Mitochondrial Fission Alleviates Zearalenone-Induced Mitochondria-Associated Endoplasmic Reticulum Membrane Dysfunction in Piglet Sertoli Cells

**DOI:** 10.3390/toxins15040253

**Published:** 2023-03-30

**Authors:** Li Ma, Chuangjiang Chen, Sirao Hai, Chenlong Wang, Sajid Ur Rahman, Wanyue Huang, Chang Zhao, Shibin Feng, Xichun Wang

**Affiliations:** 1College of Animal Science and Technology, Anhui Agricultural University, Hefei 230036, China; 2Department of Food Science and Engineering, School of Agriculture and Biology, Shanghai Jiao Tong University, Shanghai 200240, China; 3Anhui Province Engineering Laboratory for Animal Food Quality and Bio-Safety, Hefei 230036, China

**Keywords:** piglet Sertoli cells, zearalenone, mitochondria-associated endoplasmic-reticulum membrane, mitochondrial fission

## Abstract

This study aimed to investigate the effects of zearalenone (ZEA) on piglet Sertoli cell (SC)-mitochondria-associated endoplasmic reticulum (ER) membranes (MAMs) based on mitochondrial fission, and to explore the molecular mechanism of ZEA-induced cell damage. After the SCs were exposed to the ZEA, the cell viability decreased, the Ca^2+^ levels increased, and the MAM showed structural damage. Moreover, glucose-regulated protein 75 (Grp75) and mitochondrial Rho-GTPase 1 (Miro1) were upregulated at the mRNA and protein levels. However, phosphofurin acidic cluster protein 2 (PACS2), mitofusin2 (Mfn2), voltage-dependent anion channel 1 (VDAC1), and inositol 1,4,5-trisphosphate receptor (IP3R) were downregulated at the mRNA and protein levels. A pretreatment with mitochondrial division inhibitor 1 (Mdivi-1) decreased the ZEA-induced cytotoxicity toward the SCs. In the ZEA + Mdivi-1 group, the cell viability increased, the Ca^2+^ levels decreased, the MAM damage was repaired, and the expression levels of Grp75 and Miro1 decreased, while those of PACS2, Mfn2, VDAC1, and IP3R increased compared with those in the ZEA-only group. Thus, ZEA causes MAM dysfunction in piglet SCs through mitochondrial fission, and mitochondria can regulate the ER via MAM.

## 1. Introduction

Zearalenone (ZEA) is a mycotoxin with estrogenic activity, produced by the *Fusarium* species, which frequently contaminates foods and feeds [1]. It can competitively bind estrogen receptors in the uterus, mammary gland, testes, and other tissues, thus playing a toxic role [2]. In addition, ZEA can also induce miscarriages and decrease libido and sperm counts. Zearalenone shows dose-dependent toxicity, and low concentrations of ZEA can cause cell proliferation, possibly because ZEA is carcinogenic, which encourages cell carcinogenesis. Furthermore, a high concentration of ZEA can lead to cell apoptosis and oxidative stress, as well as damaging the endoplasmic reticulum (ER) and mitochondria (Mito) structures [3]. Sertoli cells (SCs) are a type of germ cell involved in the formation of the blood–testis barrier, playing an important role in spermatogenesis and sperm protection [4,5]. Testicular oxidative injury is one of the most important induction factors in male sterility. Liu et al. showed that ZEA decreased cell viability and induced goat-SC autophagy and oxidative stress [6]. Our previous study showed that ZEA damaged the structures of Mito and the ER, producing large amounts of reactive oxygen species (ROS), as well as inducing oxidative stress and the apoptosis of piglet SCs [7].

Mito and ER are two of the most important organelles in cells. The ER is responsible for protein synthesis and modification and contains a high concentration of Ca^2+^, whereas Mito provide energy for cells. These two organelles are closely related to each other in many life activities [8,9]. According to electron microscopy, the Mito-associated ER membrane (MAM) is a structure in which Mito and ER are closely associated, but keep a certain distance, of about 10 to 100 nm. [10]. The MAM is not a fixed subcellular structure, but changes depending on the state of the cells [11]. There are thousands of proteins localized in the MAM, some of which form protein complexes [12]. The maintenance of the MAM structure depends on the functional proteins and protein complexes that are localized in the MAM. These functional proteins and protein complexes also determine the function of the MAM, i.e., signal transmission, the regulation of intracellular Ca^2+^ homeostasis, and cell apoptosis [13].

Mito are highly dynamic, and they maintain their shape, size, and function via fusion and fission [14]. Mitochondrial fusion is the fusion of the outer and inner Mito membranes (IMM) of two adjacent Mito to form a single mitochondrion with a fibrous extension and a network structure, which is one of the important features of mitochondrial dynamics [15]. Mitochondrial fission is the division of a mitochondrion into two progeny of Mito, and it is mediated mainly by dynamin-related protein-1 (DRP1) [16]. A dynamic balance of mitochondrial fusion and fission is required to maintain mitochondrial homeostasis. Cells lacking DRP1 contain highly interconnected mitochondrial networks that are formed by continuous fusion in the absence of fission activity. The genetic deletion of *DRP1* resulted in the dramatic elongation of Mito in a variety of cell lines and several animal models [17].

The MAM plays an important role in controlling mitochondrial morphology and dynamics. The MAM regulates mitochondrial fission, but whether mitochondrial fission regulates the proteins and structure of the MAM in ZEA-damaged SCs is unknown.

Herein, ZEA, alone or with mitochondrial division inhibitor 1 (Mdivi-1), was used to treat piglet SCs. The effects of the ZEA on the piglet SCs’ MAM and the molecular mechanism of the ZEA-induced SC damage were investigated, to provide a basis for ZEA prevention and control.

## 2. Results

### 2.1. ZEA’s Effects on Piglet SCs’ Viability

The cell viability dropped to 50% upon exposure to 45 uM of ZEA (Figure 1A,B). When the cells were pretreated with different concentrations of Mdivi-1 followed by ZEA (45 μM), 1.5 μM Mdivi-1 resulted in the highest cell viability compared with the ZEA-alone control group (Figure 1C).

### 2.2. Piglet SCs Ca^2+^ Levels in Response to ZEA

Compared with that in the control group, the ZEA (45 μM) increased the Ca^2+^ levels significantly (*p* < 0.01; Figure 2A,B). However, the ZEA + Mdivi-1 decreased the Ca^2+^ level significantly compared with that in the ZEA group (*p* < 0.01).

### 2.3. Piglet SCs’ MAM Structures under ZEA Treatment

The control group had readily observable Mito and ER structures (Figure 3A–C). The ZEA induced significant increases in ER dilation, ER rupture, mitochondrial cristae rupture, mitochondrial vacuolization, and MAM distance (*p* < 0.01). The Mdivi-1 pretreatment significantly prevented these ZEA-induced changes (*p* < 0.01).

### 2.4. Correlation of Mito and ER of Piglet SCs in Response to ZEA

In the ZEA group, the Pearson’s correlation coefficient (PCCS) between the ER and Mito decreased significantly (*p* < 0.01) compared with that in the control group. By contrast, in the ZEA + Mdivi-1 group, the PCCS increased significantly (*p* < 0.01) compared with those in the ZEA group (Figure 4A,B).

### 2.5. Piglet SC Gene Expression in Response to ZEA

The ZEA decreased the relative mRNA expression of *IP3R* (encoding inositol 1,4,5-trisphosphate receptor), *VDAC1* (encoding voltage-dependent anion channel 1), *Mfn2* (encoding mitofusin2), and *PACS2* (encoding phosphofurin acidic cluster protein 2) significantly compared with the control group (*p* < 0.01). By contrast, the ZEA increased the relative mRNA expression of *Grp75* (encoding glucose-regulated protein 75) and *Miro1* (encoding mitochondrial Rho-GTPase 1) significantly compared with the control group (*p* < 0.01). The pretreatment with Mdivi-1 increased the IP3R, VDAC1, Mfn2 (*p* < 0.01), and PACS2 (*p* < 0.05) expression significantly, and decreased the expression of Grp75 (*p* < 0.05) and Miro1 (*p* < 0.01) significantly, compared with the ZEA-only group (Figure 5).

### 2.6. Piglet SC Protein Expression in Response to ZEA

The protein levels of PACS2, Mfn2, VDAC1, and IP3R decreased significantly in response to the ZEA compared with those in the control (*p* < 0.01). By contrast, the levels of Miro1 and Grp75 increased significantly in response to the ZEA (*p* < 0.01). These effects were largely reversed by the Mdivi-1 pretreatment compared with the ZEA-alone group (all *p* < 0.01; Figure 6).

## 3. Discussion

This study examined whether the inhibition of mitochondrial fission alleviated ZEA-induced MAM dysfunction in piglet SCs. A previous study from our laboratory showed that ZEA can damage cell structures, impair cell growth, and induce cell death and the apoptosis of SCs [7]. The present study showed that the SCs’ viability was dose-dependently decreased by the ZEA, and the cells’ viability was about 50% when they were treated with ZEA at 45 μM. After pretreatment with Mdivi-1, the cell viability increased, reaching the highest level at 1.5 μM, after which it decreased.

The homeostasis of intracellular Ca^2+^ plays an important role in cellular physiological processes [18]. The results of this study showed that the ZEA altered the Ca^2+^ homeostasis, resulting in an increase in intracellular Ca^2+^. The pretreatment with Mdivi-1 alleviated this ZEA-induced imbalance in intracellular Ca^2+^.

In our previous study, the combination of ZEA and deoxynivalenol (DON) induced the apoptosis of SCs through the ER pathway, and pretreatment with Mdivi-1 alleviated the damage to the cell ultrastructure, Mito, and ER, as well as reducing the apoptosis rate and the expression of apoptosis-related proteins and genes in the ER pathway. This indicated that Mito can be used as the upstream of ER-pathway apoptosis to participate in SC apoptosis induced by co-exposure to ZEA and DON [19]. However, the question of whether the inhibition of mitochondrial fission based on the interaction between Mito and ER affects the damage induced by ZEA in piglet SCs’ MAM has not yet been investigated. Herein, it was shown that ZEA caused structural damage to MAM, increased the MAM distance significantly, and decreased the PCCS between the ER and the Mito. Together, these two indices demonstrated that ZEA disrupts the MAM structure. The MAM is involved in Ca^2+^ regulation; therefore, the ZEA-induced MAM damage resulted in an imbalance in Ca^2+^ homeostasis. The pretreatment with Mdivi-1 restored the MAM structure. Therefore, the inhibition of mitochondrial fission can reduce the damage to SCs caused by ZEA.

There are thousands of proteins enriched in MAM that are closely associated with the formation and stability of the MAM structure, most notably Mfn2, PACS2, Miro1, and the IP3R–GRP75–VDAC1 protein complex [20]. The Mfn2 is mainly localized in the outer mitochondrial membrane (OMM) [21], and it has been reported that the overexpression of Mfn2 could ameliorate Cu-induced MAM dysfunction [22]. Therefore, Mfn2 is essential to maintain the MAM structure. The PACS2 is localized in the MAM, and the silencing of its expression inhibited the oxidized low-density-lipoprotein-induced mitochondrial localization of PACS2 and the formation of the MAM [23,24]. Furthermore, Miro1 is located in the OMM, where it mediates mitochondrial motility, contributing to the maintenance of mitochondrial morphology and function [25,26]. The IP3R–GRP75–VDAC1 complex is a MAM-protein complex, in which IP3R occurs in the ER membrane. The VDAC1 is an OMM protein. The Grp75 is mainly located in the MAM [27]. The chaperone Grp75 acts as a bridge connecting IP3R and VDAC1, and the complex comprising these three proteins can specifically transport Ca^2+^ from the ER to the Mito, as well as helping to maintain the MAM structure [28]. When Grp75 was knocked down, the functional coupling between IP3R and VDAC1 was disrupted, and Ca^2+^ transfer was reduced [29]. The abnormal expression of IP3R, Grp75, or VDAC1 leads to MAM dysfunction. In recent years, the interaction between Mito and ER has become a significant topic in cell biology. It has been reported that the treatment of cells with Mdivi-1 could reduce the apoptosis of hippocampal neuron cells induced by acquired epilepsy, as well as reducing the expression of the ER stress (ERS) markers, GRP78 and CHOP [30]. Dromparis et al. observed that ERS inhibition altered mitochondrial fission and fusion, and improved the anti-apoptosis ability of Mito [31]. However, no reports have indicated that, based on the ER-Mito interaction, ZEA damages the MAM by altering the functions of MAM proteins in SCs via the mitochondrial-fission pathway.

This study demonstrated that ZEA decreased IP3R, VDAC1, Mfn2, and PACS2 expression significantly, but increased Grp75 and Miro1 expression significantly. The pretreatment with Mdivi-1 alleviated the ZEA-induced abnormal expression of proteins and genes. The authors maintain that ZEA damaged the functional proteins that supported the MAM structure and Ca^2+^ transport, leading to an increase in the MAM distance and damage to its structure, which eventually caused the intracellular Ca^2+^ levels to rise. The pretreatment with the Mdivi-1 inhibited mitochondrial fission and alleviated the ZEA-induced abnormal protein expression in the ER membrane, OMM, and MAM. Thus, Mito and the ER are physically and functionally coupled, and Mito can regulate the ER through the MAM. Therefore, via the mitochondrial fission pathway, ZEA damages MAM functional proteins in piglet SCs.

## 4. Conclusions

Herein, the findings demonstrated that the inhibition of mitochondrial fission alleviated the damage to the MAM by ZEA. Accordingly, we hypothesized that ZEA might first interfere with the expression of related genes and proteins in the MAM, leading to structural damage, ultimately resulting in intracellular-Ca^2+^ overload. Furthermore, we demonstrated that Mito regulate the ER via MAM, which theoretically supports the prevention and control of ZEA toxicity.

## 5. Materials and Methods

### 5.1. Reagents and Chemicals

BLUEFBIO (Shanghai, China) provided the piglet SCs. The Mdivi-1 was obtained from TargetMol (Boston, MA, USA). The ZEA was purchased from Sigma Aldrich (St. Louis, MO, USA). High-glucose Dulbecco’s modified Eagle’s medium (DMEM) was obtained from Procell (Wuhan, China). The APE×BIO (Houston, TX, USA) company provided the Cell Counting Kit-8 (CCK-8) kit. Servicebio (Wuhan, China) company provided the primary antibodies used for Western blotting, together with the radioimmunoprecipitation assay (RIPA) buffer and the bicinchoninic acid (BCA) Protein Assay Kit. Solarbio (Beijing, China) provided Fluo-3-pentaacetoxymethy1 ester (Fluo-3 AM). Mitotracker^®^Deep Red FM and ER-Tracker Blue-White DPX were obtained from Maokangbio (Shanghai, China). Sigma (Darmstadt, Germany) provided the RNAse. The TRIZOL reagent was obtained from Life Technologies (Carlsbad, CA, USA) and novoprotein (Suzhou, China) provided the SYBR green qPCR mi. Sangon Biotech (Shanghai, China) synthesized the gene-specific primers.

### 5.2. Cell Culture and Treatments

Piglet SCs were cultivated in high-glucose DMEM (supplemented with 100 μg/mL streptomycin, 100 U/mL penicillin, and 10% fetal bovine serum) at 37 °C in a 5% CO_2_ incubator. ZEA (10 mg) was dissolved in 1 mL of dimethyl sulfoxide (DMSO) as a standard, and a mother solution of 5 mg Mdivi-1 standard in 1 mL DMSO was prepared and preserved at 20 °C. According to the experimental requirements, these solutions were thawed and diluted using cell-culture medium.

Primary SCs grown to 50–70% confluence were treated with ZEA (45 μM), Mdivi-1 (1.5 μM), or ZEA (45 μM) + Mdivi-1 (1.5 μM) for 24 h. Prior to ZEA exposure, Mdivi-1 was used to pretreat the cells for 3 h. The DMEM alone was used to treat the control group. Each experiment was carried out three times.

### 5.3. Cell-Viability Assay

Cells were added to 96-well plates (at 1 × 10^4^ cells per well) and grown for 24 h. Next, each well received 10 μL of CCK-8 reagent and incubation was continued for 2 h. The absorbance at 450 nm was used to calculate cell viability (Thermo Mk3, Waltham, MA, USA).

### 5.4. Determination of Ca^2+^

Cells were cultured for 24 h in 6-well plates, followed by incubation with 1 μM Fluo-3 AM at 37 °C for 30 min. The cells were collected and subjected to FACS Calibur flow cytometry (BD, Franklin Lakes, NJ, USA) to detect their Ca^2+^ fluorescence intensity. For the details, see our laboratory’s previous study [32].

### 5.5. MAM-Structure Determination

Cultured cells were digested using trypsin, followed by centrifugation and discarding of the supernatant. The cells were then fixed using 1mL 2.5% glutaraldehyde in a refrigerator at 4 °C for 24 h, followed by dehydration, embedding, and other steps. The TEM (JEM-1400, JEOL Ltd., Tokyo, Japan) was then used to assess the damage to the MAM structure. Herein, if the distance between ER and Mito was 100 nm or less, as judged using Image J (NIH, Bethesda, MD, USA), MAM is considered (*n* = 3). Each MAM was measured using three independent distances and the average was determined, as detailed by Wang et al. [33].

### 5.6. Assay for Mito-ER Correlation

Cells were treated for 24 h before being incubated in the dark for 30 min with a working solution comprising MitoTracker^®^ Deep Red FM (75 nM) and ER-Tracker Blue-White DPX (1 μM) at 37 °C. Images of the cells were then obtained under LSCM (Olympus BX51, Tokyo, Japan). The PCCS of the ER and the Mito was calculated using Image J [33].

### 5.7. Quantitative Real-Time Polymerase Chain Reaction (qRT-PCR)

The qRT-PCR was carried out as described previously [7]. The *GAPDH* (encoding glyceraldehyde-3-phosphate dehydrogenase) was used as the housekeeping gene. The primers used are shown in Table 1.

### 5.8. Western Blotting (WB)

Western blotting was carried out as reported previously [7]. The following primary antibodies were used: anti-VDAC1 (1:3000), anti-IP3R (1:3000), anti-PACS2 (1:3000), anti-Mfn2 (1:3000), anti-Miro1 (1:3000), anti-Grp75 (1:3000), and anti-β-actin (1:2000). Image J was used to determine the grayscale values of the immunoreactive protein bands.

### 5.9. Statistical Analysis

Data are shown as the mean ± standard deviation (SD) (*n* = 3). One-way analysis of variance (ANOVA) and Tukey’s test were used to compare the different groups. A *p* value less than 0.05 was recognized as having statistical significance. The histograms were constructed with the aid of GraphPad Prism version 7 (GraphPad Inc., San Diego, CA, USA). The formula for calculating cell viability was [(Absorbance of the experimental group-Absorbance of the blank group)/(Absorbance of the control group-Absorbance of the blank group)] × 100%.

## Figures and Tables

**Figure 1 toxins-15-00253-f001:**
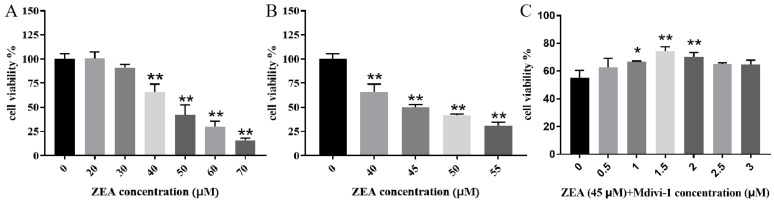
Effects of zearalenone (ZEA) alone and in combination with mitochondrial division inhibitor 1 (Mdivi-1) on cell viability of piglet Sertoli cells. (**A**,**B**) Piglet Sertoli cells’ viability under different concentrations of ZEA. (**C**) Cell viability under different concentrations of Mdivi-1 combined with ZEA (45 μM). * and ** indicate *p* < 0.05 and *p* < 0.01, respectively, compared with the control group.

**Figure 2 toxins-15-00253-f002:**
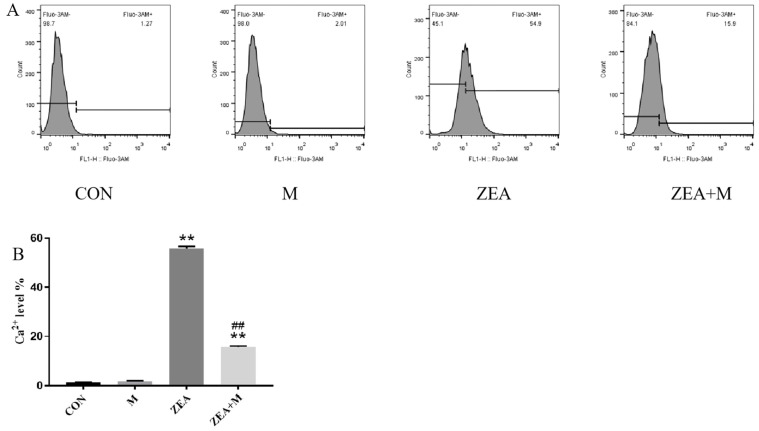
The ZEA’s effects on Ca^2+^ levels in piglet Sertoli cells. (**A**) Flow cytometry (FCM) detection of Ca^2+^ levels in the four groups. (**B**) Intracellular Ca^2+^ level. Note: CON, control group; M, Mdivi—1 group; ZEA, ZEA group; ZEA + M, ZEA+Mdivi—1 group. ** indicates *p* < 0.01 compared with the control group ## indicates *p* < 0.01 compared with the ZEA group.

**Figure 3 toxins-15-00253-f003:**
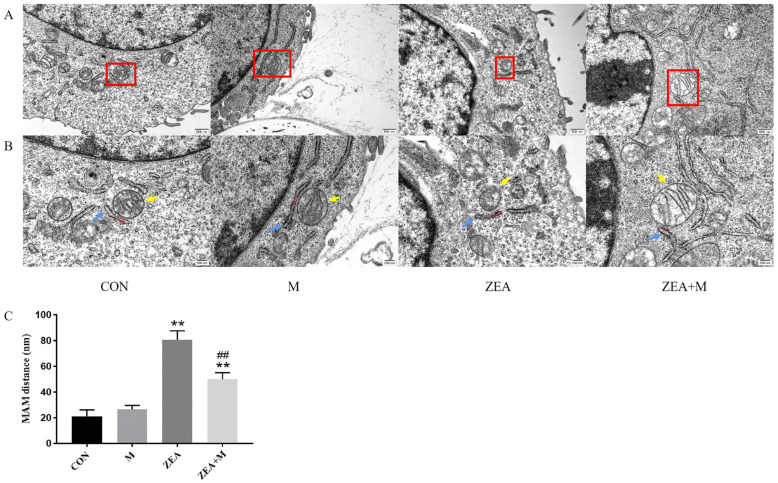
Piglet SCs’ MAM structures under ZEA treatment. Transmission-electron microscopy (TEM)-detected piglet SCs’ ultrastructure, scale bar = 500 nm (**A**) and =200 nm (**B**). (**C**) Piglet SCs’ MAM distance. Quantitative statistics were determined using the red-boxed Mito and ER. ER, blue arrows; Mito, yellow arrows; MAM, red arrows. ** indicates *p* < 0.01 compared with the control group, ## indicates *p* < 0.01 compared with the ZEA group.

**Figure 4 toxins-15-00253-f004:**
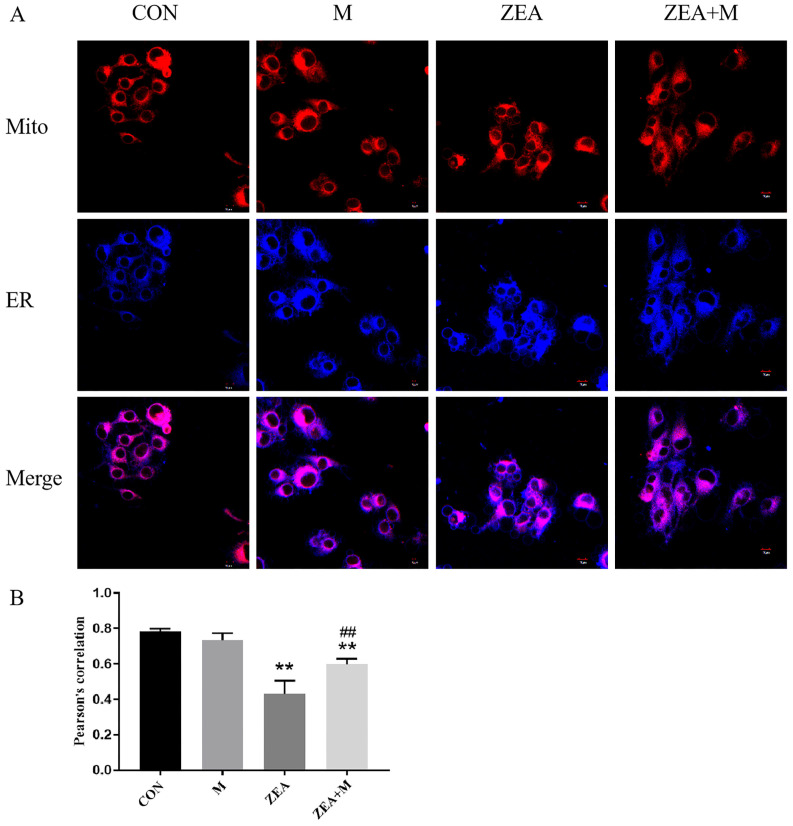
Correlation of Mito and ER of piglet SCs in response to ZEA. (**A**) The correlation between the ER and Mito was detected using laser scanning confocal microscopy (LSCM). Scale bar = 10 μm. (**B**) PCCS of ER and Mito and in piglet SCs. ** indicates *p* < 0.01 compared with the control group, ## indicates *p* < 0.01 compared with the ZEA group.

**Figure 5 toxins-15-00253-f005:**
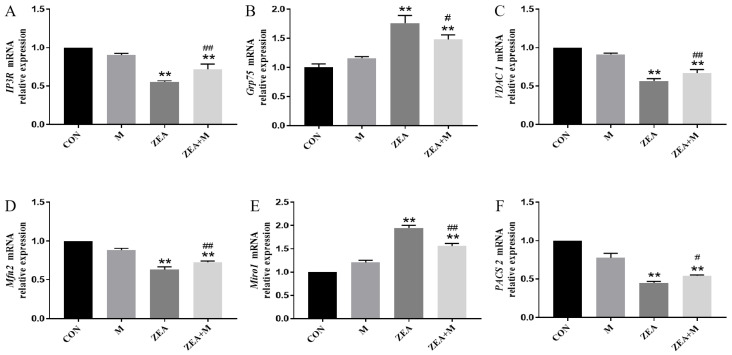
Piglet SC gene expression in response to ZEA. (**A**–**F**) The mRNA expression levels of *IP3R*, *Grp75*, *VDAC1*, *Mfn2*, *Miro1*, and *PACS2* detected using qRT-PCR. # indicates *p* < 0.05 in comparison with the ZEA-only group. ** indicates *p* < 0.01 compared with the control group, and ## indicate *p* < 0.01 compared with the ZEA group.

**Figure 6 toxins-15-00253-f006:**
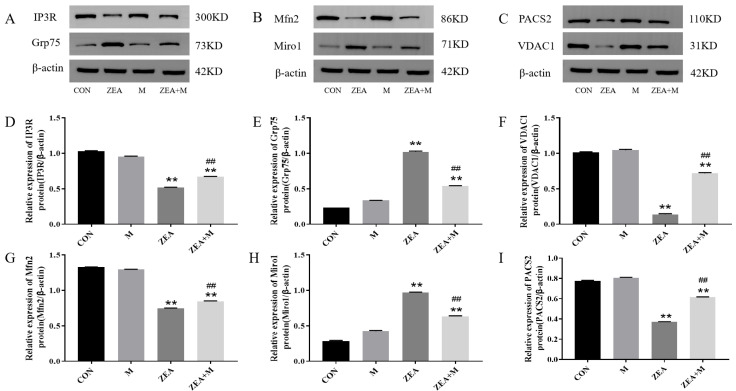
Levels of proteins in piglet SCs in response to ZEA. (**A**–**I**) Levels of IP3R, Grp75, VDAC1, Mfn2, Miro1, and PACS2 assessed using Western blotting. ** indicates *p* < 0.01 compared with the control group, ## indicates *p* < 0.01 compared with the ZEA group.

**Table 1 toxins-15-00253-t001:** Primers for different genes used in this study.

Gene	Accession Number	Primer Sequences (5′→3′)	Primer	Product/bp
*GAPDH*	396,823	TGACCCCTTCATTGACCTCC CCATTTGATGTTGGCGGGAT	F R	160
*PACS-2*	110,258,361	AGCACAGTGCAGGACAC CACTATCCCAACCTTCACGA	F R	180
*Grp75*	100,521,183	GCACGAGGAAAGCCTTAGAG GGAGAAGATGGGACGACAAA	F R	166
*Mfn2*	100,512,354	ATGGGCATTCTCGTTGTTGGAG GCAGCTTCTCGCTGGCGTAC	F R	175
*Miro1*	100,523,466	CACAAGCCTTCACTTGCAATA GTGTCACGTGCGGGTA	F R	90
*IP3R*	397,454	TTCCATCCTAACGGAACGAG CTCTGTAGTCAGCTCCTTGG	F R	127
*VDAC1*	397,010	CCCACGTATGCCGATCTT GTCAGACCATATTCGGTCCA	F R	197

## Data Availability

Not applicable.

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
