# Peer review of "Inhibition of Mitochondrial Fission Alleviates Zearalenone-Induced Mitochondria-Associated Endoplasmic Reticulum Membrane Dysfunction in Piglet Sertoli Cells"

_toxins, 2023, doi:10.3390/toxins15040253_

Round 1

Reviewer 1 Report

Review of the manuscript entitled Zearalenone induces mitochondria-associated endoplasmic reticulum membranes dysfunction in piglet Sertoli cells based on mitochondrial fission

The manuscript brings a novel data on mechanism by which ZEA induce impairment of mitochondria associated ER membranes (MAM) in piglet Sertoli cells. Inhibition of mitochondrial fission by mitochondrial division inhibitor (Mdivi-1) ameliorate ZEA-induced cytotoxicity as well as damage of MAM. Results on gene and protein expression (upregulation of glucose-regulated protein 75  (Grp75) and mitochondrial Rho-GTPase 1 (Miro1) and downregulation of inositol 1,4,5-trisphosphate receptor (IP3R), voltage-dependent anion channel 1 (VDAC1), mitofusin2 (Mfn2), and phosphofurin acidic cluster protein 2 (PACS2)) lead to conclusion that ZEA may first interfere with the expression of related  genes and proteins in MAM, damaging the structure of MAM which result with intracellular Ca2+ overload. Mdivi-1 restored the structure of MAM, ER, and mitochondria and reduced intracellular Ca2+.  

Manuscript is well organised but, in my opinion the writing of the manuscript requires improvement. Although I am not a native English speaker, I believe that English should be improved.

In general, believe that the manuscript has significant scientific potential and could be accepted for publication in Toxins after major revision.

Specific comments are listed below.

1.      Title should be changed into Inhibition of mitochondrial fission alleviates zearalenone-induced mitochondria-associated endoplasmic reticulum membranes dysfunction in piglet Sertoli cells

2.      Introduction:

-        Delete lines 30-34 (Mycotoxins are molds metabolites, including deoxynivalenol (DON), zearalenone (ZEA), aflatoxin B1 (AFB1), etc, which are widely occurs in animal feeds and crops such as corn and wheat, posing high risks for poultry, livestock, and human health [1]. ZEA is also known as an F-2 toxin, it can be metabolized into a variety of forms in animals, and is mainly present in the α-ZOL form after metabolism in pigs [2]). Instead of that I suggest: ZEA is mycotoxins with estrogenic activity, produced by Fusarium species that widely contaminate foods and feeds.

-        Delete lines 50-52 (In the 20th century, with the development of science, Ruby et al. used electron microscopy observed the extremely close contact between outer mitochondrial membrane (OMM) and endoplasmic reticulum, and was eventually named mitochondria-associated endoplasmic reticulum membranes (MAM) by Jean Vance [6]). Instead of that I suggest: According to electron microscopy MAM is a structure in which mitochondria and ER are closely associated, but keep a certain distance of about 10 and 100 nm.

-        Regarding MAM proteins in the introduction some changes should be made because authors explained the role of dynamic related protein 1 (Drp1), mentioned Mfn1 and Mfn2 and give nothing about proteins reported in the present paper (Grp75, Miro1, VDAC1…), I suggest rewriting lines 61-84 considering my comment.

3.      Results

-        Delete lines 87-88. Instead of that I suggest: Cell viability dropped to 50% upon exposure to 45 uM of ZEA.

-        Titles of all figures should be improved, give the general title of the figure (e.g. Figure 1. Effects of ZEA alone (A, B) and in combination with Mdivi-1 (C) on cell viability of piglet Sartoli cells) and after that give the legends for the particular signs or necessary explanations.

-        Results under 2.2. are not relevant for the cytotoxicity of ZEA, can be excluded or given in supplementary material

4.      Discussion

-        This part must be significantly improved, it must be devoid of redundant sentences that speak in general about the ZEA and structure of the MAM, as it was described in the introduction. Authors should focus on the discussion of gene and protein expression covered by the results. Lines 96-202 and 213-268 are the most relevant, other can be excluded.

-        According to the results of cell viability and Ca, the cytotoxicity of ZEA is the most probably linked to intracellular Ca increase, this should be emphasized after lines 96-202 and in the further text linked to the role of studied proteins explaining the possible mechanism involved in ZEA action in MAM.

-        Line 213 In our present study, the combination of ZEA and DON … should be corrected In our previous study….

5.      Conclusion

-        This part is well written except the last sentence “Our study will provide theoretical support for the prevention and control of ZEA toxicity, which will 276 further demonstrate that mitochondria can regulate ER via MAM”. It is not clear to me what the authors mean by that, maybe that study showed that Mdivi-1 plays important role in diminishing ZEA toxicity which provides new insights for possible prevention or neutralisation of ZEA toxic effects on reproductive cells.

6.      Methods

-        5.3. Detection of cell viability: I assume that authors calculated IC50 of ZEA (45 uM), how this was calculated, please describe in Statistics

-        5.4 Observation of cell growing status in not relevant and can be excluded

Reviewer 2 Report

This manuscript presents a study on the effect of zearalenone on pig mitochondria of piglet Sertoli cells.

The paper seems well constructed and to present nice figures on pathway of the proposed toxicity.

As a drug toxicity and P450 specialist, I am missing some important informations.

It would be nice to show the structure of Zearalenone and its principal metabolite.

I have reviewed in the past many papers and thesis concerning toxicity to the mitochondria. Thus I am astonished that no paper of Bernard Fromenty group is cited. May be in your former papers on oxidative stress?

The papers reads well.

I noted a few typos :

Line 251: the name is not Peter D., but Peter Dromparis

Line 318: JEOL (not JEIL)

Line 327  and 338 : In fact Image J was developed at NIH Bethesda by Wayne Rasband (NIH), Fidji Image J is a distribution with some plugins.

Thus the paper is well written, seems to be well conceived. Some more emphasis on : why this toxicity to Sertoli cells is important, should be made in the introduction.

It should be acceptable after very minor corrections

Round 2

Reviewer 1 Report

The manuscript has been significantly improved, there are still a few type errors in the text, eg line 10 more than 2x. Supplementary material should be under Cell culture and treatment, and supplementary material is not required. I recommend reviewing the English language. The manuscript can be accepted after these minor revisions
